# The Impact of the COVID-19 Pandemic on Tourists’ WTP: Using the Contingent Valuation Method

**DOI:** 10.3390/ijerph18168605

**Published:** 2021-08-14

**Authors:** Chang-Young Jeon, Hee-Won Yang

**Affiliations:** 1Department of Tourism Administration, Kangwon National University, Chuncheon-si 200-701, Korea; magic0207@kangwon.ac.kr; 2Division of Research, Research Institute for Gangwon, Chuncheon-si 244-61, Korea

**Keywords:** willingness to pay, contingent valuation method, knowledge of tourism risk, perceived risk, attitude

## Abstract

This study estimated tourists’ willingness to pay (WTP) for tourist sites or facilities in the prolonged COVID-19 pandemic by applying the dichotomous choice-contingent valuation method to two different tourism destination types. A survey was conducted among domestic tourists in South Korea who had visited destinations within the last six months. We conducted a logistic regression with 1283 effective samples. The results showed differences in tourists’ WTP, depending on type, and the factors affecting WTP differed. Tourists with higher tourism attitude and knowledge of tourism risk exhibited a higher WTP. Tourists with higher perceived risk of infectious disease exhibited less WTP.

## 1. Introduction

The tourism industry is sensitive to political changes, economic crises, and natural disasters [1] and is severely affected by infectious diseases [2,3,4,5]. Infectious diseases threatening the health of tourists influence their behaviour in selecting tourist destinations [6,7,8], and ultimately change the tourism environment and ecosystem [9].

During the COVID-19 pandemic, countries with high inflows of international tourists have been more prone to COVID-19 transmission and deaths [10]. Therefore, the global tourism industry is facing a crisis [11,12], as extreme countermeasures are being taken to control the spread of the disease by restricting travel and closing national borders [4,9,13]. Tourism demands have declined drastically due to the effects of COVID-19, including increased psychological anxiety and restricted admission to tourism facilities [9,12,14]. In fact, according to the Korea National Tourism Survey [15] report, domestic tourism in 2020 decreased significantly compared to 2019 (number of trips −35.0%, the travel days −40.9%, and the amount of spending −45.7%).

Despite the risk and anxiety associated with COVID-19, there are emerging tourist activities that reduce this risk, such as visiting relatively less crowded tourist destinations [9,16]. In fact, ‘untact’ tourism began in South Korea to meet the demand for leisure and tourism activities while minimizing the risk of infection [11]. Local governments have been promoting ‘untact’ tourist attractions in open spaces, such as hiking courses and parks, by utilising unmanned payment systems in kiosks. Various tourism venues, including hotels and resorts, have also created a culture of untact service [9,11].

Thus, it is crucial to assess changes in the tourism environment caused by the COVID-19 pandemic, which is expected to continue. Previous studies have investigated tourists’ behavioural intentions during the COVID-19 pandemic [8,11,17,18,19], particularly regarding social costs [4], travel patterns [9], and tourist profiling [20]. Further, Bae and Chang [11] highlighted the importance of untact tourism during health crises. Jeon and Yang [9] argued that a certification system to designate safe tourist destinations is needed to minimize the propagation of the virus. In addition, Sánchez-Cañizares et al. [8] indicated that tourists are willing to pay extra to ensure their safety. However, few studies have quantified tourists’ expenses, identifying the factors affecting their willingness to pay (WTP) for safe tourism during the ongoing COVID-19 pandemic.

Therefore, we used the contingent valuation method (CVM) to estimate the amount of money that tourists are willing to pay for safe tourism offerings during the COVID-19 pandemic. To minimize bias, we gauged participants’ responses using admission fees as a realistic payment method. In this study, the respondents answered two different types of surveys for tourist sites and tourism facilities. The two distinguished types were examined to compare the influencing factors and WTP for tourism facilities with safety certificates. Therefore, this study conducted a logistic regression analysis to estimate the extra fees tourists are willing to pay to avoid the risk of contracting an infectious disease. Moreover, this study attempted to identify tourists’ perceptions of tourism risk and attitudes during the COVID-19 pandemic.

Through this study, we can explore the tourist’s risk awareness and attitudes changed by COVID-19. Additionally, the impact of these factors on tourist’s WTP can be quantitatively presented. The findings of this study can be used as baseline data for local governments and tourism companies to establish policies and strategies to overcome the COVID-19 crisis. Furthermore, the findings can contribute to building an effective design that promotes the speedy recovery of the demand for tourism and the delivery of safe tourism services.

## 2. Literature Review

### 2.1. COVID-19 and Tourism

Previous social anxiety-inducing crises, such as the SARS epidemic (2002–2004) and the Middle East Respiratory Syndrome (MERS, 2012–2015), made the tourism industry suffer [9,21], but it recovered quickly [4] and continued to grow [22]. However, the COVID-19 pandemic has wreaked havoc on the global tourism industry due to the stringent measures implemented to curb the spread of the virus. This has led to a sharp decline in the number of global tourists (by 1.1 billion), tourism revenue (by 1.2 trillion USD), and jobs (by 120 million) [22]. Countries with a high inflow of international tourists have been especially affected by the COVID-19 outbreak [22].

Moreover, COVID-19 has transformed the tourism environment. The severity and infectivity of COVID-19 have caused people to fear travel, resulting in protective behaviours and travel avoidance [23]. Therefore, tourists now prefer travelling to neighbouring regions for short periods and visiting less crowded open spaces, such as natural attractions [9], avoiding high-risk places [16]. Thus, COVID-19 has completely transformed the preference and pattern of tourism activities.

Tourists’ concern about the probability of contracting an infectious disease, such as COVID-19, greatly affects their decision making regarding the selection of tourist destinations and behavioural intentions [2,6,7]. In particular, risk perception in travel has changed since the COVID-19 crisis began [14]. Sánchez-Cañizares et al. [8] identified the relationship between the perception of the travel risks and tourism behaviours during the COVID-19 pandemic. In this context, research on tourists’ risk perception and intentions is key to restoring tourism demand [8]. Therefore, many scholars have conducted tourism research in relation to COVID-19. Such studies have explored the following: behavioural intentions regarding untact tourism [11]; COVID-19’s effects on post-pandemic planned travel behaviours [17]; tourists’ intention to use hotels following the COVID-19 pandemic [18]; tourists’ intention to travel based on risk perception [8]; the impact of tourists’ risk knowledge on behavioural intention [19]; the mitigation of perceived risks to travel behaviour [24]; the estimation of social costs [4]; structural changes in local tourism networks [9]; market recovery strategies for cruise businesses [25]; tourist profiling [20]; a resilience-based framework for reviving the global tourism industry post-COVID-19 [12]; a mechanism for formulating the travel and leisure industry’s recovery strategies [26]; and a new travel risk scenario by analysing travel risk perception during the pandemic [23].

The discussions on direct COVID-19 tourism impacts, attitudes, and practices are important to achieve a recovery in the tourism industry [26]. According to Villacé-Molinero et al. [14], tourists’ type and experience do not affect their travel decisions, and risk perception regarding tourism has changed since the COVID-19 pandemic began. Additionally, the level of risk in the event of a crisis is perceived differently based on individual experience and knowledge [19,27], whereas perceived risk hurts tourism [19] and behavioural intentions [8,11,28]. Considering the relationship between behavioural intention and the actual manifestation of behavioural intention in the form of WTP [8], it is especially important to quantify the determinant factors in tourists’ WTP; however, studies on this topic are scarce.

### 2.2. Risk Perception and Tourism Attitude

Risk perception refers to subjective value determination in an uncertain situation that stems from a particular risk [29]. In tourism, risk perception is defined as ‘the probability that an action may expose tourists to danger that can influence travel decisions if the perceived danger is deemed to be beyond an acceptable level’ [2] (p. 384). The experiential and intangible attributes of tourism lead tourists to perceive a higher level of risk [30]. Risk perception is a key factor affecting the decisions of tourists [16,31,32], especially regarding the selection of tourist destinations [4,19,33].

Infectious diseases are major risk factors that have a serious impact on the tourism industry [7,34,35]. Infectious diseases spread to local communities through tourists and create chaos [36,37]. To minimize health hazards from infectious diseases, tourists practice hand washing, wear masks [7], and develop other preventive habits [38,39]. As COVID-19 is highly contagious, tourists’ risk perception is undoubtedly high [8,10,40]. Therefore, travel risk perception has changed since the COVID-19 crisis began [14].

Individual tourists’ perception of risk exerts a negative effect on their attitude as it implies their prediction of loss [8]. According to empirical studies conducted by Lobb et al. [41] and Quintal et al. [42], risk perception has a negative correlation with attitude. Further, risk perception has been found to negatively affect attitudes [11,19]. Zheng et al. [23] showed that the severity and susceptibility to the threat of infectious diseases cause travel fear. Health risk propensity has been observed to negatively affect individuals’ scores on the Pandemic Anxiety Travel Scale (PATS) [43]. In addition, constraints arising from epidemic situations create negative biases, which negatively affect behavioural intention. Other researchers have found that tourists were willing to pay extra fees for safety measures [8]. According to research conducted by Villacé-Molinero et al. [14], travel risk perception has changed since the COVID-19 crisis began, and decisions to cancel or not to cancel travel depended on government communication.

In the meantime, risk perception is closely related to tourists’ knowledge, such as tourists’ cultural dispositions [44], past experiences, similar experiences, familiarity with the event, and pursuit of curiosity [45]. Considering that risk perception originates from uncertainty, rich experiences in tourism and sufficient knowledge of tourism risk diminish tourists’ risk perception [27] and exert a positive effect on tourism attitude and behavioural intention [19].

### 2.3. Contingent Valuation Method (CVM)

For this study, we implemented the CVM, presenting tourists with a hypothetical situation to assesses their WTP for tourism resources, which have strong attributes of public goods [1]. This allowed us to quantify the economic value of the relevant resources. The CVM can be used to estimate not only use value, but also non-use value; therefore, it is often utilized in studies to estimate the value of tourism resources [46,47,48].

Open-ended or closed questioning methods are usually used to estimate an individual tourist’s WTP. Open-ended questions are easy to manage and are used to minimize the starting point bias [49]. However, open-ended questions tend to produce no response or competitive bidding, and create vast gaps among respondents [50], presenting the problem of decreased reliability of respondents’ strategic behaviours and the analysed result [51]. In contrast, closed-ended questions are usually used in dichotomous choice (DC) or iterative bidding games. In particular, DC questions have been reported to minimize the burden on respondents and to represent the preference of respondents more accurately. Thus, they have the merit of measuring WTP more accurately than other survey methods [51].

The CVM has been widely used to assess the value of a variety of tourism resources, such as cultural heritage [8,52,53], festivals [54,55,56,57], national parks [47,58], and environmental and natural resources [46,59,60].

Nevertheless, few studies have estimated tourists’ WTP in hypothetical situations where infectious diseases are prevalent. A study measured the social costs during the recent COVID-19 pandemic [4], but individual tourists’ WTP was not estimated. A study by Sánchez-Cañizares et al. [8] assessed the relationship between risk perception of COVID-19 and WTP; however, WTP was measured on a Likert scale, not based on the actual costs that tourists were willing to pay. For this reason, this study aims to examine the impact of risk perception and attitude on WTP during the COVID-19 pandemic and analyse the tourists’ WTP and the determinant factors that influence it.

## 3. Methodology

### 3.1. Survey Instrument and CVM Setup

To estimate tourists’ WTP for services that allow them to avoid risk, we designed a questionnaire survey based on criteria used in previous studies. The questionnaire comprised four sections. The first section informed the respondents about the purpose of the study, including the following information: ‘Please rest assured that the content of this survey will be kept strictly confidential according to Article 33 of the Statistical Law and collected information will be used for statistical purposes only’. The respondent was asked if they agreed to participate in this survey in advance, and only continued onto the next stage if they answered affirmatively.

The second section focused on risk perception and tourism attitudes. To measure tourist and tourism attitudes we referred to previous studies and derived three subcategories each [4,8,11,19,42]. To measure tourism attitude, we referred to Park et al. [61], So et al. [62], Bae and Chang [11], Sánchez-Cañizares et al. [8], and Zhu and Deng [19], and selected three subcategories. Each category was measured on a 7-point Likert scale (1= totally disagree, 7 = totally agree).

The third section of the questionnaire focused on tourists’ WTP. First, we asked the respondents if they planned to travel in next three months. To measure the WTP of tourists who responded ‘yes’, we used hypothetical situations [46,63]. We described the challenges imposed by COVID-19 on tourism activities, and presented participants with a hypothetical situation in which the pandemic was prolonged. Thereafter, we asked the respondents if they were willing to pay a mandatory admission fee for using tourism destinations similar to those they had visited before.

The dichotomous choice-contingent valuation method (DC-CVM) has many merits, but is prone to bias due to overestimation of the value [46]. To minimize this, we presented the bid value in the form of an ‘admission fee’, representing an actual monetary value, and surveyed the responses using two different types of tourist sites or tourism facilities. Type 1 entailed the use of pre-existing tourist sites and tourism facilities, whereas Type 2 entailed the use of tourist sites or tourism facilities with a safety certification (e.g., having an accreditation for observing and complying with a mandate for fever checks, disinfection and sanitation protocols, and other preventive measures against contagious diseases) offering untact tourism services (e.g., practices of social distancing through unmanned ticket issuing, buying tickets in advance, and admitting tourists at time intervals). Using these two distinct types of surveys, we measured tourists’ WTP for using tourist sites where safe and untact tourism services are guaranteed.

To minimize the bias on the bid value of the admission fee, we conducted the survey in two stages. First, we conducted a preliminary survey using open-ended questions (participants: n = 150) and selected 7000 KRW, 10,000 KRW, 15,000 KRW, 20,000 KRW, and 30,000 KRW as the scope of bid values based on the findings of the preliminary survey. The main survey was designed so respondents could choose either ‘yes’ or ‘no’ to the DC questions regarding their WTP the suggested bid value and visit the site.

The final section of the survey focused on respondents’ demographic characteristics, including gender, age, marital status, educational level, and average monthly income, as these factors can potentially affect tourists’ WTP [4,46,47,64,65].

### 3.2. Measurement Model

In this study, we used the DC-CVM to examine tourists’ WTP for using tourist sites and tourism facilities during the COVID-19 pandemic, as well as the determinant factors that influence WTP. To this end, we asked the respondents if they were willing to pay a randomly chosen bid value. The utility function for this is shown in Equation (1):(1)$$u_{j}=v_{j}\left(q_{j},m,x\right)+\varepsilon_{j},\;j=1,\;0$$

$v_{j}$: j = 1 when the respondent is willing to pay the admission fee suggested in the indirect utility function explained by an observable variable; j = 0 otherwise.

m: income; x: individual social and economic variables (e.g., age, education, marital status); and $\varepsilon_{j}$: unobserved, random variable (random variable with an average value of 0).

If the respondent chooses to pay the admission fee, a randomly given bid value of A KRW, and chooses ‘yes’ to use the tourist sites or tourism facilities, this can be expressed as $v_{1}\left(q_{1},m-A,x\right)$. However, there is indirect utility if the respondent chooses not to pay the admission fee, a randomly given bid value, and gives up using the tourist sites or tourism facilities; this can be expressed as $v_{0}\left(q_{0},m,x\right)$ [46,66]. Therefore, as the bid value (A KRW) changes, the change in utility can be expressed as ∆v [67]. The difference in utility for respondents’ answers under the survey of two different types is shown in function (2) [68].
(2)$$\Delta v=v_{1}\left(q_{1},\;m-A,x\right)-v_{0}\left(q_{0},m,x\right)+\left(\varepsilon_{0}-\varepsilon_{1}\right)$$

In this study, the utility difference is the continuous data of the independent variable as follows: paying A KRW and using tourist sites or tourism facilities, or not paying A KRW and giving up using the tourist sites or tourism facilities. If the dependent variable is discrete data (0 or 1), the logit model is used with the maximum likelihood estimation [46]. In this study, we used the truncated mean WTP to estimate and compare the WTP of the other two types. In particular, the truncated mean WTP is considered more appropriate due to the statistical consistency, efficiency with aggregation ability, and theoretical constraints [69]. This method estimates the value by numerical integration from 0 to Max A (maximum bid value) as function (3).
(3)$${\mathrm{WTP}}_{\mathrm{truncated}}=\int_{0}^{\mathrm{Max}\;A}f_{\eta }\left(\Delta v\right)d=\frac{1}{\beta }\ln\left(\frac{1+\exp\left(\alpha \right)}{1+\exp\left(\alpha +\beta \;\mathrm{Max}\;A\right)}\right)$$

### 3.3. Site Selection and Data Collection

The target sites of this study were limited to areas operated by local governments and natural resource-oriented tourist sites such as local or national parks, arboretum, and natural recreational forests, excluding recreational tourist attractions and tourist facilities such as museums, theme parks, and casinos. The tourist sites and tourism facilities operated by local governments are relatively cheaper than those operated by private enterprise, so there is no burden for tourists to use them. In addition, local governments are actively pursuing strategies and policies for minimizing the spread of COVID-19 and expansion of the influx of tourists, such as through mandatory checking of tourists’ body temperature, disinfection, and limiting the number of visitors. Additionally, due to the COVID-19 pandemic, tourists prefer spatially open natural resource-oriented tourism resources to spatially closed tourism facilities [9]. Thus, it is appropriate to set the research site as a natural resource-oriented tourist sites and tourist facility to analyse the tourist’s WTP and its determinants in time of prolonged COVID-19 pandemic.

Sampling was performed at two different time points. The first sampling was conducted via a preliminary survey of 150 people who had a tourism experience within the last six months. The question items asked for the admission fees at the tourist sites or tourism facilities they recently visited, and asked for the maximum amount they were willing to pay to revisit the sites or to visit similar tourist sites. Based on this preliminary survey, we derived the scope of the bid value and revised the errors in some question items before conducting the main survey. The main survey targeted only Koreans who intended to travel within the next three months. As foreign tourists were unable to enter or move into the country due to the COVID-19 pandemic, the survey was administered only to domestic tourists. The survey was conducted for one month (from 1 to 30 November 2020), via online panel research by Date Spring. The allocation sampling method was used to evenly collect the age and gender samples of the respondents, the survey was distributed to 1400 people (135 people for each bid value and for each type) and the samples were thus collected.

Among the collected samples, incomplete or incorrectly marked responses were excluded. A total of 1283 samples were included in the analysis. The response ratio per bid value for each type is presented in Table 1 and Figure 1.

## 4. Findings

### 4.1. Sample Profile

The sociodemographic information of respondents is shown in Table 2. For both types, the proportion of women was slightly larger (51.5%, 50.3%) than men (48.5%, 49.7%). Over 60% of the respondents were married and between 20 and 49 years old. Most respondents had a college education and earned 2.5–6.99 million KRW (approximately 2270–6350 USD) monthly.

### 4.2. Validity and Reliability of the Measurement Model

This study conducted a confirmatory factor analysis on each scenario type (Type 1 entailed the use of pre-existing tourist sites and tourism facilities, whereas Type 2 entailed the use of tourist sites or tourism facilities with a safety certification) to derive the construct validity of the observed variables that measure the latent variable, which influences WTP. Construct validity was determined by assessing all item loadings and their associated t-values, composite reliabilities (CR), and average variance extracted (AVE) [61,70]. Table 3 represents the results of CFA. All item loadings exceeded 0.7 and were significant at *p* < 0.001. The CR values exceeded 0.7, and the AVE values exceeded 0.5. These values met the criteria suggested by Fornell and Larker [71] and Hair et al. [70], confirming the convergent validity of the construct.

To measure discriminant validity, it was necessary to compare whether the square root of AVE of each construct was greater than the correlation value between the concerning factor and other factors [71]. The results of the analyses are presented in Table 4. The square root of the AVE of each construct is greater than the correlation value, which presents discriminant validity.

### 4.3. Determinants of Willingness to Pay

The results of binary logistic regression analysis for each type are presented in Table 5. First, in Type 1, we found that the bid value exerted a significant negative (−) effect on WTP. Meanwhile, knowledge of tourism risk, perceived risk of infectious diseases, and monthly income exerted a significant positive (+) effect on WTP. Specifically, knowledge of tourism risks and monthly income had a relatively large impact on WTP.

In contrast, in Type 2 (tourism destination with safety certificates), it was found that bid value, perceived risk of infectious disease, and age exerted a significant negative (−) effect on WTP. Conversely, it was found that knowledge of tourism risk, tourism attitude, and monthly income exerted a significant positive (+) effect on WTP. Among these factors, knowledge of tourism risk and attitude had a greater impact on WTP than other factors.

The factors that significantly influenced WTP varied slightly for each type. In particular, bid value, knowledge of tourism risk, perceived risk of infectious disease, and monthly income exerted a significant effect on both types. Tourism attitude and age were found to be significant factors in Type 2 only. Meanwhile, education and marital status did not exert any effect on either type.

These results are consistent with previous studies’ findings that the COVID-19 outbreak has changed tourist perceptions and attitudes [14], and that increased risk recognition for infectious diseases has a negative impact on attitudes and behaviours [8,11,19]. In addition, the type and experience of tourists are, in part, consistent with prior research [14] showing that these factors do not affect travel decisions. However, the results of prior research should be considered alongside the knowledge that the rich experience and knowledge of tourists reduce risk recognition, positively affecting attitudes and behaviour [19,27]. Therefore, discussions on tourists’ risk awareness and experience are important for the rapid recovery of the tourism industry in line with the prolonged COVID-19 pandemic.

Calculating the truncated mean WTP of an individual tourist by performing a binary logistic regression analysis showed the following result. The average WTP was 10,830 KRW (approximately 9.8 USD) for Type 1 and 18,923 KRW (approximately 17.1 USD) for Type 2 (Figure 2). This shows that the respondents were willing to pay an average of 8093 KRW (approximately 7.3 USD) more for tourist sites and tourism facilities with safety certification and untact services.

Therefore, in consideration of the study [14] showing that communication with the government plays an important role in tourists deciding whether to cancel travel, the government should provide accurate information about infection and risk during the COVID-19 pandemic to help tourists make the right decisions. These results can help the researchers to better understand tourist behaviour and take concrete steps to help the tourism sector recover amid the crisis caused by COVID-19.

## 5. Discussion and Conclusions

### 5.1. Theoretical Implications

This study has the following theoretical implications. First, WTP was mainly affected by bid value, education, and monthly income, even though there was a slight variation based on survey type. Within the context of the ongoing pandemic, for this study, we included variables such as knowledge of tourism risk, perceived risk of infectious diseases, and tourism attitude. Nevertheless, bid value, education, and monthly income variables remain important variables. This supports previous studies’ findings, which showed that higher bid values negatively affected tourists’ WTP [4,46,58,72]. The determinant factors that positively affected WTP also matched those of previous studies, namely, income [46,47,64,65,73,74] and education [58,65,67].

Second, WTP is often affected by risk perceptions and attitudes. In this study, a higher level of knowledge of tourism risks exerted a positive effect on WTP, which is an actual manifestation of behavioural intention [8]. This is consistent with previous studies, which reported that a rich tourism experience and sufficient knowledge of tourism risks can diminish risk perception and positively affect tourism attitude and behavioural intention [19,27]. Meanwhile, the perceived risk of infectious diseases negatively affected WTP. This supports previous study findings that risk perception of infectious disease negatively affects attitude and behavioural intention [8,11,19]. This clearly shows that tourists’ knowledge and risk perception are important research topics in the field of tourism as key factors influencing tourists’ behavioural intentions.

Third, there was a difference between the two types in terms of individual average WTP. Specifically, tourists’ WTP for safe and untact tourism in Type 2 was higher by 8093 KRW (approximately 7.3 USD). This supports the findings of Sánchez-Cañizares et al. [8], who examined WTP for higher safety measures when travelling during the COVID-19 pandemic. Furthermore, this also highlights the need for safety and untact tourism in the future [9,11,75]. Therefore, to develop sustainable tourism strategies in the future, it will be important to conduct research on changes in tourism behaviours by considering the outbreak of infectious diseases.

### 5.2. Practical Implications

This study has a few practical implications. First, according to Gossling et al. [75] and Sánchez-Cañizares et al. [8], COVID-19 will have lasting effects on the tourism industry, changing its landscape. In this study, knowledge of tourism risks had a positive impact on WTP. Due to heightened tourist anxiety during the COVID-19 pandemic, it is necessary to aid tourists’ decision making by providing them with sufficient information on infectious diseases and tourism risks. According to Zheng et al. [23], coping and resilience have a positive effect on cautious travel. Therefore, constantly providing tourists with accurate information on safety measures and possible risks will allow for safe tourism even within the context of a pandemic. This could contribute to the rapid recovery of the tourism industry.

Currently, tourism venues in Korea check tourists’ temperature and disinfect the facilities constantly. However, each tourist destination has its own protocols. Therefore, local governments can prepare safety instruction manuals for tourist destinations and operate safe tourist sites by accrediting tourist destinations. This accreditation system could be a foundation to mitigate the effects of infectious diseases in the long run.

### 5.3. Limitations and Suggestions for Future Studies

This study has the following limitations. First, this study’s generalizability is limited as its participants were all Korean. Therefore, future studies need to improve the representativeness of their samples by enrolling participants from other countries.

Second, this study presented risk perception by dividing it into knowledge of tourism risk and the perceived risk of infectious diseases. However, we did not compare tourists’ nationality, destination-specific cultural characteristics, and differences in personal disposition. Considering that tourists’ goals are diverse, and the tourism market is becoming segmented, future studies should focus on presenting specific tourism activities to meet tourists’ demands.

Third, cross-sectional data were acquired via a questionnaire survey. Although we included questions about WTP, we did not follow up on actual tourism activities and payment status. To establish a strong causal relationship and to improve the quality and effectiveness of the findings, it is imperative to examine the connection with actual tourism behaviours.

Therefore, future research should consider these restrictions. It is necessary to conduct research through a national comparative study, or by establishing data that can track actual tourism activities. These studies will be the basis for rapid response in the event of a recurrence of pandemic situations, such as that with COVID-19.

This study is meaningful in pandemic situations as it assessed tourists’ WTP for extra fees for safety; furthermore, it explored the factors that affect WTP. The results of this study are expected to contribute to the rapid recovery of the tourism industry and markets at a time when the pandemic situation is expected to be prolonged.

## Figures and Tables

**Figure 1 ijerph-18-08605-f001:**
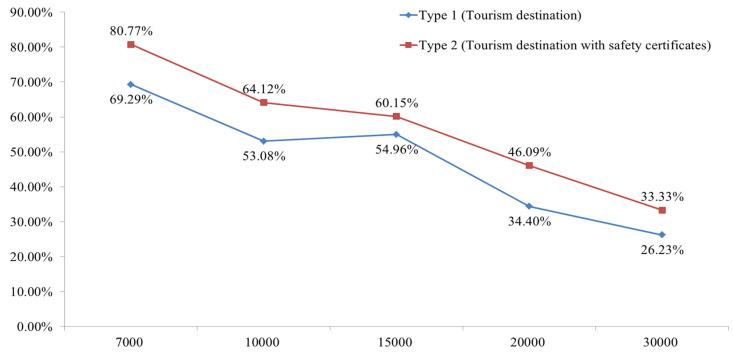
Probability of ‘yes’ response on WTP. Note: X axis is bid value (KRW), Y axis is the percentage of ‘yes’ responses.

**Figure 2 ijerph-18-08605-f002:**
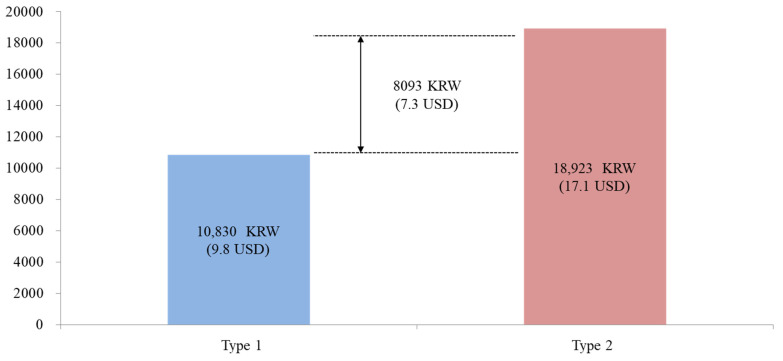
WTP for 1 person.

**Table 1 ijerph-18-08605-t001:** Response ratio per bid value.

Bid Value (KRW)	Type 1		Type 2	
	Yes (%)	No (%)	Yes (%)	No (%)
7000	98 (77.2%)	29 (22.8%)	109 (83.8%)	21 (16.2%)
10,000	79 (60.8%)	51 (39.2%)	101 (77.1%)	30 (22.9%)
15,000	86 (65.6%)	45 (34.4%)	98 (73.7%)	35 (26.3%)
20,000	62 (49.6%)	63 (50.4%)	76 (59.4%)	52 (40.6%)
30,000	42 (34.4%)	80 (65.6%)	61 (48.4%)	65 (51.6%)
Total	367 (57.8%)	268 (42.2%)	445 (68.7%)	203 (31.3%)

Note: 1000 KRW = 0.91 USD.

**Table 2 ijerph-18-08605-t002:** Sociodemographic profile of respondents (N = 1283).

Variable (Unit)	Categories	Type 1 (n = 635)		Type 2 (n = 648)	
		n	%	n	%
Gender	Male	308	48.5	322	49.7
	Female	327	51.5	326	50.3
Age	Below 20	105	16.5	110	17.0
	20–29	129	20.3	131	20.2
	30–39	132	20.8	135	20.8
	40–49	124	19.5	127	19.6
	50–59	79	12.4	75	11.6
	60 and older	66	10.4	70	10.8
Marriage status	Single	222	35.0	254	39.2
	Married	413	65.0	394	60.8
Education	Middle school and lower	25	3.9	28	4.3
	High school diploma	119	18.7	138	21.3
	University	405	63.8	393	60.6
	Master’s or doctoral degree	86	13.5	89	13.7
Monthly income (Million KRW)	Below 250	33	5.2	39	6.0
	250–399	118	18.6	132	20.4
	400–549	220	34.6	191	29.5
	550–699	110	17.3	132	20.4
	700–849	72	11.3	76	11.7
	850–999	45	7.1	42	6.5
	More than 1000	37	5.8	36	5.6

**Table 3 ijerph-18-08605-t003:** Results of validity and reliability analyses.

Factors and Items		Mean	SD	Alpha	CR	AVE
Type 1	KT1. I know about the causes of tourism risks	5.127	1.101	0.851	0.788	0.553
	KT2. I know about the consequences of tourism risks	5.085	0.901			
	KT3. I know about the solutions to tourism risks	4.937	0.844			
	PR1. I know about the harm caused by infectious diseases	5.164	1.219	0.822	0.811	0.588
	PR2. I know about the current affected range of infectious diseases	5.122	1.127			
	PR3. I know about the preventive measures for infectious diseases	5.072	0.965			
	TA1. Tourism is valuable	5.182	0.912	0.815	0.824	0.609
	TA2. Tourism is beneficial	5.109	0.881			
	TA3. Tourism is attractive	5.065	0.877			
Type 2	KT1. I know about the causes of tourism risks	5.598	0.943	0.881	0.801	0.573
	KT2. I know about the consequences of tourism risks	5.348	0.918			
	KT3. I know about the solutions to tourism risks	5.204	0.882			
	PR1. I know about the harm caused by infectious diseases	5.198	1.021	0.852	0.823	0.608
	PR2. I know about the current affected range of infectious diseases	5.013	1.017			
	PR3. I know about the preventive measures for infectious diseases	4.986	0.965			
	TA1. Tourism is valuable	5.327	0.964	0.839	0.808	0.584
	TA2. Tourism is beneficial	5.298	0.898			
	TA2. Tourism is attractive	5.261	0.842			

Note: All loadings of the reflective measurement model exceeded 0.7, which was significant (*p* < 0.001). SD: standard deviation; Alpha: Cronbach’s alpha; CR: composite reliability; and AVE: average variance extracted. Type 1: $x^{2}$/df = 2.013, CFI = 0.872, NNFI = 0.907, GFI = 0.964, RMSEA = 0.031, and SRMR = 0.0257. Type 2: $x^{2}$/df = 1.841, CFI = 0.915, NNFI = 0.913, GFI = 0.970, RMSEA = 0.018, and SRMR = 0.0171.

**Table 4 ijerph-18-08605-t004:** Discriminant validity analysis based on the Fornell–Larcker Criterion.

Measure	Type 1			Type 2		
	KT	PR	TA	KT	PR	TA
Knowledge of tourism risk (KT)	**0.744**			**0.757**		
Perceived risk of infectious diseases (PR)	0.518	**0.767**		0.524	**0.780**	
Tourism attitude (TA)	0.502	−0.462	**0.781**	0.427	−0.352	**0.764**

Note: The bold numbers on the diagonal are the square root of AVE. Off-diagonal numbers are the correlations among constructs.

**Table 5 ijerph-18-08605-t005:** Estimated WTP and coefficients.

	Type 1		Type 2	
Variable	Coeff.	Wald	Coeff.	Wald
Bid value	−0.00058	17.902 ***	−0.00037	37.481 ***
Knowledge of tourism risk	0.54623	5.785 *	0.76623	8.756 **
Perceived risk of infectious diseases	−0.29215	1.790 *	−0.45215	6.125 *
Tourism attitude	0.66112	7.215	0.71112	8.155 **
Age	−0.34756	2.245	−0.37456	2.216 *
Education	0.31221	1.387	0.30217	1.125
Marital status	0.21547	0.895	0.51547	1.547
Monthly income	0.51479	5.775 *	0.41479	4.549 ***
Constant	−0.12542	21.486	−0.08254	16.488
Log-likelihood	−214.024		−315.786	
Likelihood ratio statistic	41.215		62.421	
Pseudo R2	0.0559		0.07542	

Note: * *p* < 0.05, ** *p* < 0.01, and *** *p* < 0.001.

## Data Availability

The data are not publicly available due to confidentiality reasons.

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
