# Peer review of "The Impact of the COVID-19 Pandemic on Tourists’ WTP: Using the Contingent Valuation Method"

_ijerph, 2021, doi:10.3390/ijerph18168605_

Round 1

Reviewer 1 Report

Dear authors

Please, find here some suggestions and comments about your manuscript. I consider that it is a good contribution with practical implications to managers and to the tourism sector. However,  there are some improvements that must be done.

1. Abstract

Please, indicate that the survey was answered by people from Korea.

2. Introduction

Please, add more literature related to COVID-19 and the impact on Korean tourism sector. 

3. Literature review

There is a lack of literature focused on COVID-19 recently published. Please, update your references and introduce articles from the WOS (JCR, Q1 and Q2). It can improve your paper a lot and give you the opportunity to adopt a critical approach. 

Some suggestions:

Sharma, G. D., Thomas, A., & Paul, J. (2021). Reviving tourism industry post-COVID-19: A resilience-based framework. Tourism management perspectives, 37, 100786.

Villacé-Molinero, T., Fernández-Muñoz, J. J., Orea-Giner, A., & Fuentes-Moraleda, L. (2021). Understanding the new post-COVID-19 risk scenario: Outlooks and challenges for a new era of tourism. Tourism Management, 86, 104324.

Abbas, J., Mubeen, R., Iorember, P. T., Raza, S., & Mamirkulova, G. (2021). Exploring the impact of COVID-19 on tourism: transformational potential and implications for a sustainable recovery of the travel and leisure industry. Current Research in Behavioral Sciences, 2, 100033.

4. Results

I suggest to introduce a discussion about the results, constructing them with the literature. I found interesting your results, but the approach is not critical enough. It can be improved by introducing a contrast with literature published about this topics.

General

Please, check the order of your references during the text. 

Author Response

Thank you for your meticulous review. We have revised the paper based on your comments.

We would appreciate it if you would check the article again and evaluate it for publication in the International Journal of Environmental Research and Public Health.

Once again, thank you for your sincere feedback.

Point 1: (Abstract) Please, indicate that the survey was answered by people from Korea.

Response 1: We have revised the abstract according to your comment.

Point 2: (Introduction) Please, add more literature related to COVID-19 and the impact on Korean tourism sector.

Response 2: In our previous version of the manuscript we had mentioned the impact of Korea tourism due to COVID-19, citing previous research (Bae & Chang, 2021; Jeon & Yang, 2021). However, based on your advice, we decided that that was insufficient, and using data recently released by the Ministry of Culture, Sports and Tourism, we have added information about the decrease in domestic Korean tourism.

Point 3: (Literature review) There is a lack of literature focused on COVID-19 recently published. Please, update your references and introduce articles from the WOS (JCR, Q1 and Q2). It can improve your paper a lot and give you the opportunity to adopt a critical approach.

Some suggestions:

Sharma, G. D., Thomas, A., & Paul, J. (2021). Reviving tourism industry post-COVID-19: A resilience-based framework. Tourism management perspectives, 37, 100786.

Villacé-Molinero, T., Fernández-Muñoz, J. J., Orea-Giner, A., & Fuentes-Moraleda, L. (2021). Understanding the new post-COVID-19 risk scenario: Outlooks and challenges for a new era of tourism. Tourism Management, 86, 104324.

Abbas, J., Mubeen, R., Iorember, P. T., Raza, S., & Mamirkulova, G. (2021). Exploring the impact of COVID-19 on tourism: transformational potential and implications for a sustainable recovery of the travel and leisure industry. Current Research in Behavioral Sciences, 2, 100033.

Response 3: Thank you for comment and suggestions. We have supplemented our literature review by referring to the previous research you suggested.

Point 4: (Results) I suggest to introduce a discussion about the results, constructing them with the literature. I found interesting your results, but the approach is not critical enough. It can be improved by introducing a contrast with literature published about this topics.

Response 4: We agree that this section was lacking. Therefore, we have supplemented the results section according to your advice.

Point 5: (General) Please, check the order of your references during the text.

Response 5: We checked again according to your comment. Thank you again for your sincere comment.

Reviewer 2 Report

  1. The article concerns an interesting issue that is rarely addressed (in such a methodological approach), and contains cognitively original content and research results.
  1. The research method (CVN, DC-CVM) that was implemented has certain limitations, which the author(s) him/herself draws attention to in the final section of the work, but it seems that in this case it would be difficult to make use of a different, decidedly better method. In my opinion, an essential issue in the CVN method is the quality of research tools (survey) as well as the survey process itself. In the description of the research method, there is no information on how the questionnaires had been shared and distributed, and it is not explained why only domestic tourists were surveyed, without taking into account foreign tourists at all. We do not find out at all how, where and when the questionnaire was given to respondents, which makes it difficult to interpret the research. How, when and where the questionnaires are made available and distributed should be clearly indicated and justified.
  2. I regard the means of presenting the research results as competent – it is clear and consistent with the other sections of the work.
  3. The work is of great practical (applied) value and can make a significant contribution to the discussion on the recovery of the tourism industry in the post-pandemic period, especially in its first stage).
  4. I have no comments. I would like to offer my congratulations on the completion of this interesting and valuable research.

Author Response

Thank you for your meticulous review. We have revised the paper based on your comments.

We would appreciate it if you would check the article again and evaluate it for publication in the International Journal of Environmental Research and Public Health.

Once again, thank you for your sincere feedback.

Point 1: The article concerns an interesting issue that is rarely addressed (in such a methodological approach), and contains cognitively original content and research results.

Response 1: Thank for your comment and positive evaluation. Through the revision process, we hope to improve the completeness of the research.

Point 2: The research method (CVN, DC-CVM) that was implemented has certain limitations, which the author(s) him/herself draws attention to in the final section of the work, but it seems that in this case it would be difficult to make use of a different, decidedly better method. In my opinion, an essential issue in the CVN method is the quality of research tools (survey) as well as the survey process itself. In the description of the research method, there is no information on how the questionnaires had been shared and distributed, and it is not explained why only domestic tourists were surveyed, without taking into account foreign tourists at all. We do not find out at all how, where and when the questionnaire was given to respondents, which makes it difficult to interpret the research. How, when and where the questionnaires are made available and distributed should be clearly indicated and justified.

Response 2: We have supplemented the parts you mentioned (site selection and data collection) based on your advice.

Point 3: I regard the means of presenting the research results as competent – it is clear and consistent with the other sections of the work.

Response 3: Thank you for the positive review.

Point 4: The work is of great practical (applied) value and can make a significant contribution to the discussion on the recovery of the tourism industry in the post-pandemic period, especially in its first stage).

Response 4: We agree that the results of this study are tangible and that they can help the tourism industry recover during the prolonged COVID-19 pandemic.

Point 5: I have no comments. I would like to offer my congratulations on the completion of this interesting and valuable research.

Response 5: Once again, we appreciate your sincere review and comments.

Reviewer 3 Report

The introduction consists of too many paragraphs.   Title: Ambiguous what impact COVID19 has. The area of tourism is also too wide.   The analysis results and the study title do not match well. The title is very macroscopic, but the study actually looks at the difference in WTPs according to individuals' psychological conditions.   Introduction: The contribution of research is also very superficial. Based on the results of the study, there should be more detail on what can be done by using the study results.   Methodology: I think it's done properly.   The most important thing to supplement is the justification of this study. Why should this research be conducted? What can we gain? Does this research contribute to society? The authors should be able to answer the most basic questions.

Author Response

Thank you for your meticulous review. We have revised the paper based on your comments.

We would appreciate it if you would check the article again and evaluate it for publication in the International Journal of Environmental Research and Public Health.

Once again, thank you for your sincere feedback.

Point 1: The introduction consists of too many paragraphs.

Response 1: We respect your comment that the introduction consists of too many paragraphs. However, we think that the introduction is important to provide the background and necessity of this study. We compared our introduction to that of other papers published in this journal, and found similar lengths. We think that the length is appropriate, and we ask for your understanding.

Point 2: Title: Ambiguous what impact COVID19 has. The area of tourism is also too wide. The analysis results and the study title do not match well. The title is very macroscopic, but the study actually looks at the difference in WTPs according to individuals' psychological conditions.

Response 2: Thank for your recommendation about the title. We have changed the title of this article based on your comment.

Point 3: Introduction: The contribution of research is also very superficial. Based on the results of the study, there should be more detail on what can be done by using the study results.

Response 3: We supplemented the literature review based on your comment. We have also provided details on this topic in the “Discussion and Conclusions.”

Point 4: Methodology: I think it's done properly. The most important thing to supplement is the justification of this study. Why should this research be conducted? What can we gain? Does this research contribute to society? The authors should be able to answer the most basic questions.

Response 4: We understand your comment. The purpose of this study is to investigate the perception of tourists who have changed due to the COVID-19 pandemic, and to estimate the resulting tourist's WTP. This study may not live up to your expectations. However, We think it is very important and necessary to study the recovery and operation of the tourism industry at a time when infectious diseases such as COVID-19 are prevalent.

Also, we think the research is important - not just because of COVID-19, but because there have been unprecedented changes and your research can give insight into how people change in such drastic situations. There may not be another pandemic, but there will certainly be more crises, and this information can help policy-makers when it comes to determining how to help the tourism industry recover from all sorts of crises.

We understand your concern, and we have added information to the conclusion section demonstrating this study's contributions, as well as information to the introduction section explaining the necessity of the research.

Round 2

Reviewer 1 Report

Thank you for improving the paper considering the suggestions and comments made by the reviewers.

Reviewer 3 Report

The points were properly reflected.